# AutoAL: Automated Active Learning with Differentiable Query Strategy Search

## Abstract

As deep learning continues to evolve, the need for data efficiency becomes increasingly important. Considering labeling large datasets is both time-consuming and expensive, active learning (AL) provides a promising solution to this challenge by iteratively selecting the most informative subsets of examples to train deep neural networks, thereby reducing the labeling cost. However, the effectiveness of different AL algorithms can vary significantly across data scenarios, and determining which AL algorithm best fits a given task remains a challenging problem. This work presents the first differentiable AL strategy search method, named **AutoAL**, which is designed on top of existing AL sampling strategies. AutoAL consists of two neural nets, named SearchNet and FitNet, which are optimized concurrently under a differentiable bi-level optimization framework. For any given task, SearchNet and FitNet are iteratively co-optimized using the labeled data, learning how well a set of candidate AL algorithms perform on that task. With the optimal AL strategies identified, SearchNet selects a small subset from the unlabeled pool for querying their annotations, enabling efficient training of the task model. Experimental results demonstrate that AutoAL consistently achieves superior accuracy compared to all candidate AL algorithms and other selective AL approaches, showcasing its potential for adapting and integrating multiple existing AL methods across diverse tasks and domains.

## 1 Introduction

With the development of deep learning (DL) techniques, large volumes of high-quality labeled data have become increasingly crucial as the model training processes are usually data hungry Ren et al. (2021); Zhan et al. (2022). Active learning (AL) addresses this challenge by iteratively selecting the most informative unlabeled samples for annotation, thereby boosting model efficiency Xie et al. (2021); Wang et al. (2023). Deep active learning strategies are typically divided into two categories: uncertainty-based and representativeness/diversity-based approaches Sener & Savarese (2017); Zhu & Bento (2017); Ash et al. (2019). Uncertainty-based methods focus on querying samples the model is most uncertain about, while diversity-based strategies aim to select a diverse subset of samples that effectively represent the entire dataset Citovsky et al. (2021). Both strategies have limitations, especially in batch selection and varying dataset complexities, leading to the inconsistency of the strategy efficiency Ren et al. (2021). Hybrid strategies, which balance uncertainty and diversity Zhan et al. (2021a;b), offer partial solutions. However, these strategies rely on strategy choices and subjective experiment settings, which can be ineffective in real-life applications.

Previous works Baram et al. (2004); Hsu & Lin (2015); Pang et al. (2018) have explored selecting among different AL strategies to achieve optimal selections without human effort. SelectAL Hacohen & Weinshall (2024) introduces a method for dynamically choosing AL strategies for different deep learning tasks and budgets by estimating the relative budget size of the problem. It involves evaluating the impact of removing small subsets of data points from the unlabeled pool to predict how different AL strategies would perform. However, this strategy depends on approximating the reduction in generalization error on small subsets, which would not fully capture the complexities of real-world tasks. Zhang et al. (2024) proposes to select the optimal batch of AL strategy from hundreds of candidates, aiming to maximize future cumulative rewards based on noisy past reward observations of each candidate algorithm. However, both the computational cost and the lack of differentiability make the optimization of these works inefficient.

To address these challenges, we propose **AutoAL**, an automated active learning framework that leverages differentiable query strategy search. AutoAL offers two key advantages: 1) it enables automatic and efficient updates during optimization, allowing the model to adapt seamlessly; 2) it is inherently data-driven, leveraging the underlying data distribution to guide the AL strategy selection. However, integrating multiple AL strategies into the candidate pool makes achieving differentiability difficult. To overcome this issue, AutoAL leverages a bi-level optimization framework that smoothly integrates and optimizes different AL strategies. We design two neural networks: SearchNet and FitNet. FitNet is trained on the labeled dataset to adaptively yield the informativeness of each unlabeled sample. Guided by FitNet's task loss, SearchNet is efficiently optimized to accurately select the best AL strategy. Instead of searching over a discrete set of candidate AL strategies, we relax the search space to a continuous domain, allowing SearchNet to be optimized via gradient descent based on FitNet's output. The gradient-based optimization, which is more data-efficient than traditional black-box searches, enables AutoAL to achieve state-of-the-art performance in strategy selection with significantly reduced computational costs. Our key contributions are as follows:

- We introduce a novel AutoAL framework for differentiable AL query strategy search. To the best of our knowledge, AutoAL is the first automatic query strategy search algorithm that can be trained in a differentiable way, achieving highly efficient updates of query strategy in a data-driven manner.

- AutoAL is built upon existing AL strategies (i.e., uncertainty-based and diversity-based). By incorporating most AL strategies into its search space, AutoAL leverages the advantages of these strategies by finding the most effective ones for specific AL settings, objectives, or data distributions.

- We evaluate AutoAL across various AL tasks. Given the importance of AL in domain-specific tasks, particularly where data quality and imbalance are concerns, we conducte extensive experiments on both nature image datasets and medical datasets. The results demonstrate that AutoAL consistently outperforms all candidate strategies and other state-of-the-art AL methods across these datasets.

## 2  RELATED WORK

**Active Learning.**    Active Learning has been studied for decades to improve the model performance using small sets of labeled data, significantly reducing the annotation cost Cohn et al. (1996); Settles (2009); Zhan et al. (2021b). Most AL research focuses on pool-based AL, which identifies and selects the most informative samples from a large unlabeled pool iteratively. Pool-based AL sampling strategies are generally categorized into two primary branches: diversity-based and uncertainty-based. The diversity-based methods select the batch of samples that can best represent the entire dataset distribution. CoreSet Sener & Savarese (2017) selects a batch of representative points from a core set, which is a subsample of the dataset that effectively serves as a proxy for the entire set. In contrast, uncertainty-based methods focus on selecting the samples with high uncertainty, which is due to the data generation or the modeling/learning process. Bayesian Active Learning by Disagreements (BALD) Gal et al. (2017) chooses data points that are expected to maximize the mutual information between predictions and model posterior. Meta-Query Net Park et al. (2022) trains an MLP that receives one open-set score and one AL score as input and outputs a balanced meta-score for sample selection in the open-set dilemma. Yoo & Kweon (2019) incorporates a loss prediction strategy by appending a compact parametric module designed to forecast the loss of unlabeled inputs relative to the target model. This is achieved by minimizing the discrepancy between the predicted loss and the actual target loss. It requires subjective judgment and strategic selection of active learning methods to be effective on specific datasets in real-world settings.

**Adaptive Sample Selection in AL.**    Neither diversity-based nor uncertainty-based methods are perfect in all data scenarios. Diversity-based methods tend to perform better when the dataset contains rich category content and large batch size, while uncertainty-aware methods typically perform better in opposite settings Zhan et al. (2021b); Citovsky et al. (2021). Several past works have explored adaptive selection in active learning to solve this problem. A common approach is to choose the best AL strategy from a set of candidate methods and use it to query unlabeled data, such as Hacohen & Weinshall (2024) and Zhang et al. (2024) introduced in Section 1. Active Learning By Learning (ALBL) Hsu & Lin (2015) adaptively selects from a set of existing algorithms based on their estimated contribution to learning performance. It frames each candidate as a bandit machine, converting the problem into a multi-armed bandit scenario. However, none of these methods leverage

the differentiable framework for automatic AL selection. Compared to the methods mentioned above, AutoAL first relaxes the search space to be continuous, thus can optimize automatically via gradient descent. This framework enables AutoAL to seamlessly integrate multiple existing AL strategies and rapidly adapt to select the optimal strategy based on the labeled pool. The simple gradient-based optimization is much more data-efficient than traditional black-box searches. Therefore, AutoAL can significantly reduce computational costs compared to these methods.

**Bilevel Optimization.** Bilevel optimization is a hierarchical mathematical framework where the feasible region of one optimization task is constrained by the solution set of another optimization task Liu et al. (2021). It has gained significant attention due to its nested problem structure, which allows the two tasks to optimize jointly. Bilevel optimization adapts to many DL applications Chen et al. (2022), such as hyperparameter optimization Chen et al. (2019); MacKay et al. (2019), meta-knowledge extraction Finn et al. (2017), neural architecture search Liu et al. (2018); Hu et al. (2020), and active learning Sener & Savarese (2017). For instance, Sener & Savarese (2017) formulates AL as a bi-level coreset selection problem and designs a 2-approximation method to select a subset of data that represents the entire dataset. In this work, we novelly formulate the Al query strategy search as a bi-level optimization problem, i.e., iteratively updating FitNet and using its task loss to guide the optimization of SearchNet. We further reformulate it as a differentiable learning paradigm to enable a more efficient selection of examples across various objectives and data distributions.

## 3 METHODOLOGY

### 3.1 PROBLEM SETTING

Pool-based AL focuses on selecting the most informative data iteratively from a large pool of unlabeled independent and identically distributed (*i.i.d.*) data samples until a fixed budget is exhausted or the expected model performance has been reached. Specifically, in AL processes, assuming that we have an initial seed labeled set $L = \{(x_j, y_j)\}_{j=1}^{M}$ and a large unlabeled data pool $U = \{x_i,\}_{i=1}^{N}$, where $M \ll N$, $y_i \in \{0, 1\}$ is the class label of $x_i$ for binary classification and $y_i \in \{1, ..., p\}$ for multi-class classification. In each iteration, we select a batch of the most informative data $Q^* = \arg\max_{x \in U}^{b} \alpha(x, \Omega)$ with batch size $b$ from $U$ based on the basic learned model $\Omega$ and the acquisition function $\alpha(x, \Omega)$, and query their labels from oracle/annotator. With these samples, the expected loss function $f$ of the basic learned model can be minimized. $L$ and $U$ are then updated.

The performance of existing AL methods relies on the choice of query strategies, which should be carefully adapted to different tasks or applications. To achieve a robust and adaptive AL performance, this work novelly creates an automated AL strategy selection framework. AutoAL integrates two neural nets, FitNet $\Omega_F$ and SearchNet $\Omega_S$ upon each basic learned model $\Omega$ to select the best AL strategy before selecting $Q$. To facilitate automatic selection, AutoAL incorporates them into a bi-level optimization framework and relaxes the search space to enable differentiable updates. We discuss more details of AutoAL framework in the following sections.

### 3.2 AUTOMATED ACTIVE LEARNING

We propose Automated Active Learning (AutoAL), which aims to adaptively deliver the optimal query strategy for each sample in a given unlabeled data pool. Specifically, AutoAL consists of two neural networks: FitNet $\Omega_F$ and SearchNet $\Omega_S$. $\Omega_S$ selects the optimal AL strategy from a set $\mathcal{A}$, which contains $K$ candidate AL sampling strategies $\{\mathcal{A}_\kappa\}_{\kappa \in [K]}$ (e.g., BALD, Maximum Entropy, etc.). $\Omega_F$ models the data distribution within the unlabeled dataset and guides the training of SearchNet $\Omega_S$. Since annotations for the unlabeled data cannot be accessed directly by the model $\Omega$, AutoAL requires *no* data from the unlabeled pool $U$, using only the labeled dataset $L$ for training.

In AutoAL, each AL iteration consists of $\mathcal{C}$ cycles. In each cycle $c$, the labeled data $L$ is randomly split into two equal-sized subsets: training and validation sets. FitNet $\Omega_F$ computes the cross-entropy loss $\mathcal{L}$ Zhang & Sabuncu (2018) for data. Meanwhile, SearchNet $\Omega_S$ generates scores for the samples, which are used to rank them. This raises a key question: If only the training set is available, without access to the unlabeled pool, can $\Omega_S$ still effectively select the optimal samples?

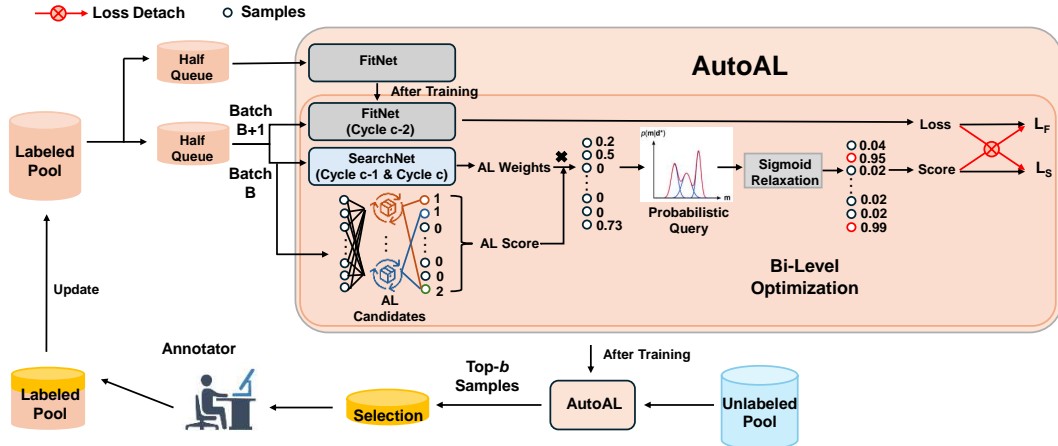

Figure 1: Overall framework of differentiable query strategy search for automated active learning (AutoAL). AutoAL leverages labeled pool to train the FitNet and SearchNet in a bi-level optimization mode. Data samples with the largest search score are then selected from the unlabeled pool.

Our key design is intuitive: SearchNet $\Omega_S$ treats the training batch as the labeled pool and the validation batch as the unlabeled pool. This allows $\Omega_S$ to simulate the process of AL selection, without direct access to the actual unlabeled data pool. The output of $\Omega_S$ is shown in Figure 1, a sample-wise score that aggregates the candidate AL strategy selection. The output of $\Omega_S$ is the informative loss of each sample. The optimization goal of $\Omega_S$ is to select data with higher losses, as they are expected to provide more informative updates to the neural network. For $\Omega_F$, the training objective is to minimize the loss of the selected samples, particularly those *prioritized* by $\Omega_S$. AutoAL is formulated as a bi-level optimization problem, as the optimization of $\Omega_F$ is dependent only on itself, but the optimization of $\Omega_S$ depends on the optimal $\Omega_F$.

$$\Omega_S^* = \arg\max_{\Omega_S} \sum_{j=1}^{M/2} \mathcal{L}_S((x_j, y_j), \Omega_S, \Omega_F^*), \tag{1}$$

$$\text{s.t.} \quad \Omega_F^* = \arg\min_{\Omega_F} \sum_{j=1}^{M/2} \mathcal{L}_F((x_j, y_j), \Omega_F).$$

$\mathcal{L}_F$ and $\mathcal{L}_S$ represent the losses of $\Omega_F$ and $\Omega_S$, respectively. The neural nets $\Omega_F$ and $\Omega_S$ are optimized jointly as outlined in Eq. 1. At the lower level of the nested optimization framework, $\Omega_F$ is optimized to approach the distribution of the dataset. At the upper level, $\Omega_S$ is optimized to output the informativeness of each data, based on the optimal distribution returned from $\Omega_F$.

### 3.3 DIFFERENTIABLE QUERY STRATEGY OPTIMIZATION

Although Eq. 1 shows an effective framework for automatically deriving the optimal query strategy, solving the bi-level optimization problem is inefficient in practice. To address this, we derive a probabilistic query strategy coupled with a differentiable optimization framework, improving the efficiency of the optimization process.

**Probabilistic query strategy.** During training, for each labeled data $x_j$, we query a score $\mathcal{S}\kappa(x_j)$ from each candidate AL sampling strategy $\mathcal{A}i$, where $\mathcal{S}\kappa(x_j) \in 0, 1$ indicates whether the sample is selected. To model the overall score $\mathcal{S}(x_j)$, which is a combination of all the AL search scores $\mathcal{S}\kappa(x_j)$, we adopt a *Gaussian Mixture* approach Reynolds et al. (2009):

$$p(\mathcal{S}) = \sum_{k=1}^{k} \pi_k \mathcal{N}(\mathcal{S} \mid \mu_k, \Sigma_k), \tag{2}$$

$$\{\hat{\pi}_k, \hat{\mu}_k, \hat{\Sigma}_k\} = \arg\max_{\pi_k, \mu_k, \Sigma_k} \prod_{j=1}^{M} p(\mathcal{S}(x_j)), \tag{3}$$

where $\pi_k$ is the weight of the $k$-th gaussian component and $\mathcal{N}$ is the probability density function. Then we generate samples according to the Gaussian Mixture model and get the $t$-th maximum value, where $t$ is the ratio of batch size $b$ to the total pool size $M + N$. $W_{\kappa,j}$ is the probability of each query strategy for each sample predicted from the neural network $\Omega_S$.

$$S_{\text{sample}} \sim \sum\nolimits_{k=1}^{K} \hat{\pi}_k \, \mathcal{N}(\mathcal{S} \mid \hat{\mu}_k, \hat{\Sigma}_k), \tag{4}$$

$$\hat{\mathcal{S}}_\kappa(x_j, \Omega_S) = (\mathcal{S}_\kappa(x_j) - \vartheta_t(S_{\text{sample}})) * W_{\kappa,j} . \tag{5}$$

A sample with a higher score $\hat{\mathcal{S}}$ indicates that it has been selected by more candidate AL strategies, and therefore should be given higher priority for labeling.

**Differentiable acquisition function optimization.** If we limit the individual query score $\mathcal{S}_\kappa(x_j)$ to discrete values $\{0, 1\}$, the bi-level optimization objective of Eq. 1 becomes non-differentiable and still difficult to optimize. To enable differentiable optimization, we relax the categorical selection of strategies into a continuous space, we apply a *Sigmoid* function over all possible strategies.:

$$\bar{\mathcal{S}}(x_j) = \sum_{\kappa \in K} \frac{\lambda}{1 + \exp(-\Theta^{(j)}_{\mathcal{S}'_\kappa})} \hat{\mathcal{S}}_\kappa(x_j, \Omega_S), \tag{6}$$

where the strategy mixing weights for each sample $j$ is parameterized by a vector $\Theta^{(j)}$ of dimension $|\mathcal{A}|$. $\lambda$ is a scaling vector. $\bar{\mathcal{S}}(x_j)$ is the final differentiable learning objective function, which can be optimized with back-propagation. Therefore, $\Omega_F$ and $\Omega_S$ can be efficiently optimized under Eq. 1.

### 3.4 LEARNING AND ALGORITHM OF AUTOAL

With differentiable query strategy optimization in Sec. 3.3, we reformulate the bi-level optimization problem in Sec. 3.2 as efficient optimization objectives as follows:

$$\mathcal{L}_F = \frac{1}{B} \sum_{j'=1+(n+1)B}^{(n+2)B} \bar{\mathcal{S}}_{detach}(x'_j) \cdot \mathcal{L}(x'_j, y'_j) + \bar{\lambda}\mathcal{L}_{re}(t, B), \tag{7}$$

$$\mathcal{L}_S = -\frac{1}{B} \sum_{j=1+nB}^{(n+1)B} \bar{\mathcal{S}}(x_j) \cdot \mathcal{L}_{detach}(x_j, y_j) - \bar{\lambda}\mathcal{L}_{re}(t, B), \tag{8}$$

As described in section 3.2, in cycle $c$-2, $\Omega_F$ is optimized to get the data distribution. The final loss is calculated according to Eq. 7. In cycle $c$-1, AutoAL optimizes $\Omega_S$ to get the data sample with highest loss. The loss function from $\Omega_F$ is detached to avoid updating itself, and the final loss is calculated according to Eq. 8. Here we use negative loss to achieve gradient ascent. For $n \in \left\{0, 1, \ldots, \left\lfloor \frac{M}{2B} \right\rfloor \right\}$, AutoAL additionally integrates a loss prediction module according to Yoo & Kweon (2019) to help update $\Omega_S$, as

$$\mathcal{L}_{re}(t, B) = \left( \frac{1}{1 + \exp\left(0.5 \cdot |\alpha - t \cdot B|\right)} - 0.5 \right). \tag{9}$$

$\mathcal{L}_{re}$ represents a regularization loss that limits the number of selected samples, as shown in Eq. 9. Here, $\alpha$ is the number of selected samples, $t$ is the ratio of the query batch size $b$ in each AL iteration to the total pool size $M + N$, and $\bar{\lambda}$ is a scaling vector. The final learning objective is to select the most informative samples while effectively leveraging the underlying data distribution.

The AutoAL algorithm is illustrated in Algorithm 1. In each AL iteration, the labeled set is first used to train the FitNet and SearchNet as described in section 3.2. Then the optimal SearchNet $\Omega_S^*$ is applied to the unlabeled pool to select a new batch of data, where $\Omega_S^*$ is used to generate score $\bar{\mathcal{S}}$ for each sample $x_i$, and $Q^* = \arg\max_{x \in U}^b \bar{\mathcal{S}}(x_i)$ with top-$b$ highest scoring samples are selected and labeled. The labeled set $L$ and the unlabeled set $U$ are updated as $L = L + Q^*$ and $U = U - Q^*$, respectively. The updated labeled set is then used for task model training.

---

**Algorithm 1** AutoAL: Automated Active Learning with Differentiable Query Strategy Search

---

**Input:** $K$ candidate algorithms $\mathcal{A} = \{\mathcal{A}_\kappa\}_{\kappa \in [K]}$, labeled pool $L = \{(x_j, y_j)\}_{j=1}^M$ and unlabled pool $U = \{x_i\}_{i=1}^N$, total number of AL rounds $\mathcal{R}$, batch size $b$, task model $\mathcal{T}$.
**Output:** task model $\mathcal{T}$
 1: Initialize $\mathcal{T}$
 2: **for** $c$=1,...,$\mathcal{R}$ **do**
 3:   Optimize $\Omega_F^*$, $\Omega_S^*$ according to Eq. 1;
 4:   Calculate $\bar{\mathcal{S}}(x_i)$ by Eq. 6 for all $i \in N$;
 5:   Select the top $b$ samples with the highest score $\bar{\mathcal{S}}(x_j)$;
 6:   Query label $y_i$ for all $i \in b$; Update $L$ and $U$;
 7:   Train task model $\mathcal{T}$ by using $L$;
 8: **end for**
 9: return $\mathcal{T}$

---

Table 1: A summarization of datasets used in the experiments. The Imbalance Ratio (IR) is the ratio of the number of images in the majority class to the number of images in the minority class. IR = 1 represents balance dataset.

|  | DataSet | Training Size | Test Size | Class Numbers | Imbalance Ratio |
|---|---|---|---|---|---|
| **Nature Images** | Cifar-10 | 50,000 | 10,000 | 10 | 1.0 |
|  | Cifar-100 | 50,000 | 10,000 | 100 | 1.0 |
|  | SVHN | 73,257 | 26,032 | 10 | 3.0 |
|  | TinyImageNet | 100,000 | 10,000 | 200 | 1.0 |
| **Medical Images** | OrganCMNIST | 12,975 | 8,216 | 11 | 5.0 |
|  | PathMNIST | 89,996 | 7,180 | 9 | 1.6 |
|  | TissueMNIST | 165,466 | 47,280 | 8 | 9.1 |

## 4 EXPERIMENTS

We empirically evaluate the performance of AutoAL on a wide range of datasets with both nature and medical images. These experiments demonstrate that AutoAL outperforms the baselines by fully automating the training within the differentiable bi-level optimization framework. Additionally, we conduct ablation studies to analyze the contribution of each module and examine the effects of the candidate AL strategies, including different numbers of candidates.

### 4.1 EXPERIMENT SETTINGS

**Datasets and Baselines.** We conduct AL experiments on seven datasets: Cifar-10 and Cifar-100 Krizhevsky et al. (2009), SVHN Netzer et al. (2011), TinyImageNet Le & Yang (2015) in the nature image domain, and OrganCMNIST, PathMNIST, and TissueMNIST from MedMNIST database Yang et al. (2023) in the medical image domain. We compare our method against multiple AL sampling strategies, including Maximum Entropy Sampling Shannon (1948), Margin Sampling Netzer et al. (2011), Least Confidence Wang & Shang (2014), KMeans Sampling Ahmed et al. (2020), Bayesian Active Learning by Disagreements **(BALD)** Gal et al. (2017), Variation Ratios **(VarRatio)** Freeman (1965) and Mean Standard Deviation **(MeanSTD)** Kampffmeyer et al. (2016). We also consider the state-of-the-art deep AL methods as baselines, including Batch Active learning by Diverse Gradient Embeddings **(BADGE)** Ash et al. (2019), Loss Prediction Active Learning **(LPL)** Yoo & Kweon (2019), Variational Adversarial Active Learning **(VAAL)** Sinha et al. (2019), Core-set Selection **(Coreset)** Sener & Savarese (2017), Ensemble Variance Ratio Learning **(ENSvarR)** Beluch et al. (2018), Deep Deterministic Uncertainty **(DDU)** Mukhoti et al. (2023). We also include **ALBL** Hsu & Lin (2015) to compare AutoAL with existing selective AL strategies.

**Implementation Details.** For $\Omega_F$ and $\Omega_S$, we build the backbone using ResNet-18 He et al. (2016). We also employ ResNet-18 as the classification model on all baselines and AutoAL for fair comparison. Since AutoAL is built upon existing AL strategies and focuses on selecting the optimal strategy, we integrate seven AL methods into AutoAL: Maximum Entropy, Margin Sampling,

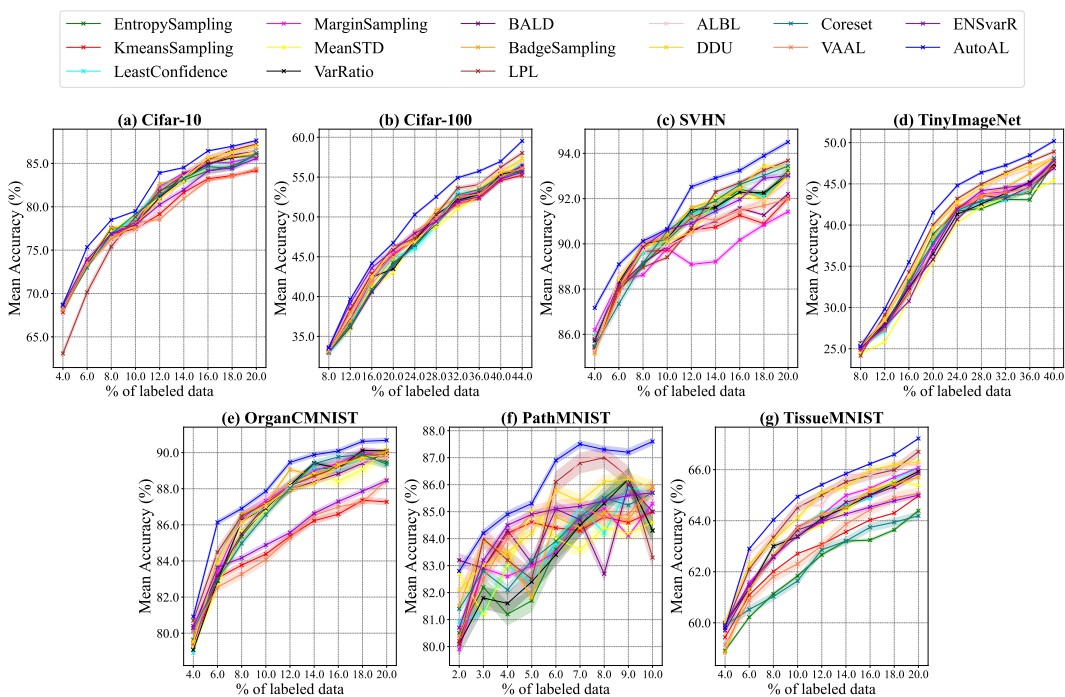

Figure 2: Overall performance on seven benchmark datasets: natural image datasets (top) and medical image datasets (bottom).

Least Confidence, KMeans, BALD, VarRatio, and MeanSTD. For the optimization of $\Omega_S$ and loss prediction module, we use SGD optimizer Ruder (2016). For the optimization of $\Omega_F$, we use Adam optimizer Kingma (2014). Both use $0.005$ as the learning rate. While training, FitNet will first update for 200 epochs using the validation queue, then $\Omega_F$, $\Omega_S$ and the loss prediction module will update iteratively with a total of $400$ epochs. All experiments are repeated *three times* with different randomly selected initial labeled pools, reporting mean and standard deviation.

### 4.2 Evaluation of Active Learning Performance

Fig. 2 shows the overall performance comparison of different AL methods, where AutoAL consistently outperforms the baselines across all datasets. Additionally, we have the following observations:

1. Our method consistently outperforms the other approaches in terms of rounds, a crucial attribute for a successful active learning method. This adaptability is particularly valuable in real-world applications, where the labeling budget may vary significantly across different tasks. For instance, one might only have the resources to annotate 10% of the data, rather than 20% or 40%.

2. Our method demonstrates robust performance not only on easier datasets but also on more challenging ones, such as Cifar-100, OrganCMNIST and TinyImageNet. Cifar-100 and TinyImageNet has significantly more classes than other datasets, while OrganCMNIST has smaller data pool, with only $12,975$ images. These challenges make active learning more difficult, yet our method's superior performance across these datasets highlights its robustness and generalizability.

3. The labeled data vs. accuracy curves of our method are relatively smooth compared to those of other methods. The baseline methods sometimes show significant accuracy drops during certain active learning rounds. This mainly dues to harmful data selection Koh & Liang (2017) or overfitting problem. However, by selecting the optimal strategy thus selecting the informative data, AutoAL can alleviate the problem and make the curve much more smooth than baseline strategies.

4. Our method shows smaller standard deviations, indicating that it is robust across repeated experiments. This robustness is crucial for AL applications, especially in a real-world setting, where the AL process is typically conducted only once.

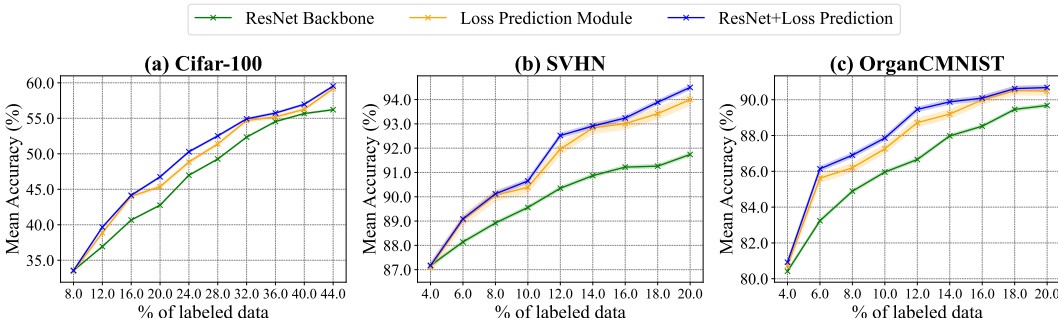

Figure 3: Ablation study on three components of AutoAL. 'ResNet Backbone': without the loss prediction module, updating only the ResNet. 'Loss Prediction Module': without updating the ResNet backbone, optimizing solely with the loss prediction module. 'ResNet+Loss Prediction': the full AutoAL pipeline.

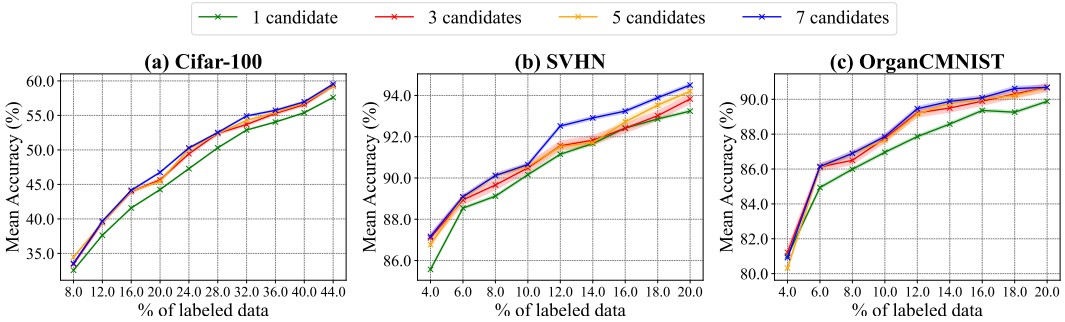

Figure 4: Ablation study on the size of the candidate pool in AutoAL strategy selection.

5. Different AL strategies produce varying results across datasets. For instance, Margin Sampling underperforms on the SVHN dataset, while KMeans Sampling and VAAL yield the worst performance on the OrganCMNIST and Cifar-10 datasets. This variability highlights the significance of our approach, which aims to identify and select the optimal strategy for each dataset.

## 4.3 ABLATION STUDIES

We perform the ablation study on two key components of AutoAL: the SearchNet architecture and the AL candidate strategies. Experiments are conducted on three datasets: Cifar-100, SVHN, and OrganCMNIST, which are chosen for their mix of natural and medical images, as well as the increased difficulty they present for active learning applications.

**SearchNet Architecture.**   Our SearchNet consists of two main components: a ResNet-18 backbone and a loss prediction module for sample loss prediction  Yoo & Kweon (2019). In this ablation study, we disable one component at a time and record the results. Fig. 3 illustrates the performance, revealing the effectiveness of both ResNet-18 backbone and the loss prediction module. However, when only the loss prediction module is used to optimize the network, performance drops significantly. This aligns with our intuition, as although the loss prediction module can assist with network optimization, the main focus shifts from selecting the best AL strategy to merely minimizing the sample loss. This misalignment will result in incorrect AL strategy weight estimation (See Fig. 1).

**Size of Active Learning Candidate Pool.**   Since AutoAL is built on multiple AL strategies, we study how the size of the AL candidate pool impacts performance. We start from the Maximum Entropy as the sole candidate for the initial baseline, then expand the candidate pool to three strategies by adding Margin Sampling and Least Confidence. For the pool of five candidates, we further include KMeans and BALD. As shown in Fig. 4, for all three datasets, the single candidate variation performs the worst but still outperforms the Entropy Sampling baseline in Fig. 2. This is because

AutoAL does not directly query data samples with maximum entropy but queries the data with the highest score from SearchNet, which has been optimized with the help of the loss prediction module and FitNet. For Cifar-100 and OrganCMNIST, AutoAL can perform well when there are at least three candidates in the selection pool. However, for SVHN, more AL candidates result in better performance. This indicates that the upper bound of candidate pool size varies across different datasets. In certain datasets, incorporating more active learning strategies yields better results, but this is not the case for other datasets. Additionally, pools with more candidates exhibit lower standard deviations. This demonstrates strong capability of AutoAL in selecting the best active learning strategy. It can consistently choose the optimal one and exclude the less effective options.

## 4.4 AL SELECTION STRATEGY

Here, we examine which AL strategy is prioritized by AutoAL in different active learning rounds. For each AL strategy candidate, we calculate the AL score of each image according to Eq. 6. Notice that we don't sum all the scores according to the dimension of each active learning, but compute the average value of all images over a round, and normalize the results for visualization. As illustrated in Fig. 5, in both experiments, KMeans dominate the sample selection during the initial rounds, such as the first round, but be de-prioritized in subsequent rounds. This indicates that diversity-based measures, which select samples that represent diverse regions of the data distribution, are more critical in the early stages of active learning. In the early rounds, when only a small percentage of the data has been labeled, the model's understanding of the entire data distribution is limited. As a result, selecting samples that cover a broad spectrum of the dataset becomes crucial to provide the model with a more comprehensive view of the data.

On Cifar-100 dataset, both Least Confidence and MeanSTD, two uncertainty-based measures, demonstrate improved performance in the final rounds. AutoAL assigns them higher priority to enhance overall performance. As the AL process progresses and more diverse data become available, the model attempts to develop a better understanding of the underlying structure of the target data. Uncertainty-based measures are more effective now because the major challenge switches to refining the decision boundaries. The transition from diversity to uncertainty-based strategies is consistent with the model's evolving needs. In early rounds, the model requires diversity to build a broad understanding of the data, but once that foundation is established, its focus shifts toward uncertainty, which helps fine-tune model decision-making ability. This adaptive approach ensures that the model targets the most diverse subsets at each stage of the learning process, from broad exploration in the early rounds to fine-grained exploitation in the later rounds. By automatically adjusting the priorities of these strategies, AutoAL improves the overall accuracy and shows stronger robustness.

## 4.5 COMPLEXITY ANALYSIS

In this subsection, we perform additional analysis experiments to showcase AutoAL's efficiency and generalizability. We mainly use the average running time to verify the results, and all the experiments

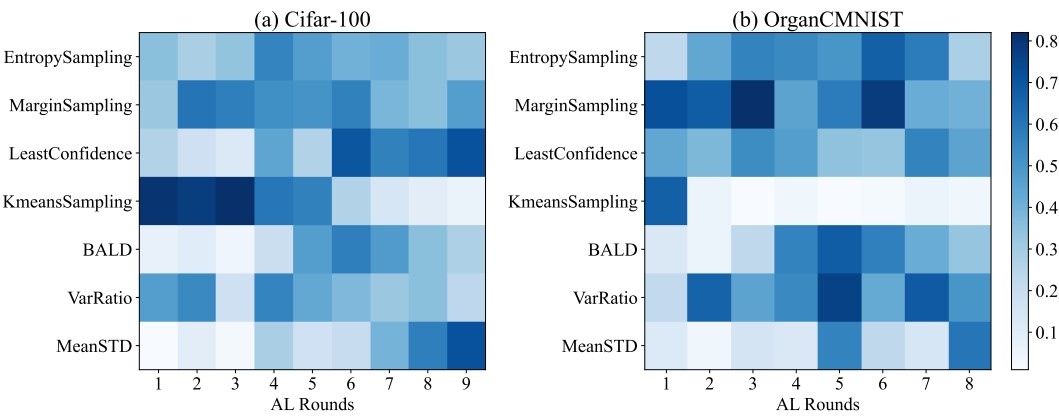

Figure 5: AL strategy scores across different AL rounds on CIFAR-100 and OrganCMNIST datasets.

Table 2: The average runtime of the entire active learning (AL) process (including training and querying), where the runtime of EntropySampling is the **unit** time. We separate total time cost of AutoAL into three main parts: candidate ALs querying AL score for images (AutoAL-ComputeALScore), SearchNet and FitNet updating (AutoAL-Search), and querying samples and training classification model after AutoAL search (AutoAL-TrainResNet).

| | CIFAR-10 | CIFAR-100 | SVHN | TinyImageNet | OrganCMNIST | PathMNIST | TissueMNIST |
|---|---|---|---|---|---|---|---|
| EntropySampling | 1.00 | 1.00 | 1.00 | 1.00 | 1.00 | 1.00 | 1.00 |
| MarginSampling | 0.99 | 1.02 | 5.97 | 0.99 | 2.01 | 1.01 | 1.84 |
| LeastConfidence | 0.85 | 1.01 | 1.03 | 0.96 | 1.85 | 0.97 | 1.81 |
| KMeansSampling | 1.20 | 1.24 | 1.59 | 0.62 | 1.33 | 0.93 | 2.37 |
| MeanSTD | 1.46 | 1.18 | 0.89 | 0.23 | 2.07 | 0.85 | 2.35 |
| VarRatio | 1.40 | 1.17 | 0.87 | 0.87 | 1.87 | 0.78 | 2.30 |
| BALD | 1.48 | 1.18 | 0.90 | 0.92 | 2.04 | 1.02 | 1.86 |
| BadgeSampling | 1.83 | 4.09 | 2.34 | 8.32 | 1.21 | 1.49 | 2.52 |
| LPL | 2.03 | 0.87 | 0.98 | 0.37 | 1.87 | 1.35 | 1.77 |
| ALBL | 1.08 | 0.57 | 1.01 | 0.82 | 2.10 | 0.88 | 1.42 |
| DDU | 1.48 | 1.87 | 2.13 | 1.62 | 1.32 | 2.32 | 1.68 |
| Coreset | 1.68 | 1.42 | 1.07 | 1.34 | 1.25 | 1.47 | 2.05 |
| VAAL | 1.24 | 1.27 | 7.02 | 2.05 | 5.28 | 4.36 | 7.68 |
| ENSvarR | 4.21 | 4.52 | 6.58 | 12.27 | 8.33 | 7.63 | 10.46 |
| AutoAL-ComputeALScore | 2.34 | 2.03 | 2.80 | 3.40 | 3.79 | 3.24 | 1.74 |
| AutoAL-Search | 0.12 | 0.11 | 0.15 | 0.18 | 0.20 | 0.17 | 0.09 |
| AutoAL-TrainResNet | 1.57 | 1.36 | 1.88 | 2.28 | 2.55 | 2.18 | 1.15 |
| AutoAL-Total | 4.03 | 3.50 | 4.83 | 5.86 | 6.54 | 5.59 | 2.95 |

was done on one Nvidia A100 GPU. We use the running time of EntropySampling as a benchmark and calculate other ALs running costs relative to it. We find:

1. Our AutoAL demonstrates its ability to generalize to large datasets, and the time cost is compatible to some of the baselines such as ENSvarR.

2. We separate the total time cost of our method into three main parts, and the AL strategy sampling will cost the most time and the update of SearchNet and FitNet will nearly cost none. This means AutoAL will time cost rely heavily on the candidate pool. When the candidate ALs are fast enough, AutoAL will be in low cost and complexity.

3. We conducted the experiment on changing the candidate pool size, and we found that when there are 3 candidates, the time cost is just 1.3x compared to EntropySampling, and our method with three candidates can also gain a high improvement on labeling accuracy. This will give real-world users flexibility in choosing the number of candidate ALs considering both accuracy and complexity.

## 5 CONCLUSION

In this paper, we have introduced AutoAL, the first automated search framework for deep active learning. AutoAL employs two neural networks, SearchNet and FitNet, integrated into a bilevel optimization framework to enable joint optimizations. To efficiently solve the bi-level optimization problem, we have proposed a probabilistic query strategy that relaxes the search space from discrete to continuous, enabling a differentiable and data-driven learning paradigm for AutoAL. This framework not only accelerates the search process but also ensures that AutoAL can generalize across a wide range of tasks and datasets. Furthermore, AutoAL is flexible, allowing for easy integration of most of the existing active learning strategies into the candidate pool, making it adaptable to evolving active learning techniques. Our extensive empirical evaluation, conducted across both natural and medical image datasets, has demonstrated the robustness and generalizability of AutoAL. The results show that AutoAL consistently outperforms baselines, highlighting its capability to optimize sample selection and enhance model performance. In future, we plan to apply AutoAL to boarder machine learning tasks such as structured prediction and consider more sophisticated AL method combinations.

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
