# OpenReview forum: "AutoAL: Automated Active Learning with Differentiable Query Strategy Search"
_ICLR.cc/2025/Conference — Submitted to ICLR 2025_

### Official Review · Reviewer_U39s · 2024-10-30

**Soundness:** 3
**Presentation:** 3
**Contribution:** 3
**Rating:** 6
**Confidence:** 4

**Summary:**

This paper attempts to tackle the "generalization problem" of active learning (AL) algorithms across data scenarios. I believe this is a core issue in the current active learning field. This paper proposes AutoAL, a differentiable AL strategy search method to select the most effective AL sampling strategies in each iteration. It consists of two neural nets, named SearchNet and FitNet, which are optimized concurrently under a differentiable bi-level optimization framework. The experiments on multiple datasets validate the effectiveness of the proposed approach.

**Strengths:**

1. The problem studied in this paper is valuable. This paper presents the first differentiable AL strategy search method.
2. The proposed AutoAL approach is interesting and easily followed.
3. The paper is well organized.
4. The experiments validate the effectiveness of the proposed approach, and the ablation study in Figure 5 is insightful.

**Weaknesses:**

1. My only concern is the efficiency of the AutoAL algorithm. Although more efficient solutions have been proposed to solve second-order optimization problems, I cannot find any relevant experiments to verify them.

**Questions:**

See the Weaknesses.

**Details Of Ethics Concerns:**

1. My only concern is the efficiency of the AutoAL algorithm. Although more efficient solutions have been proposed to solve second-order optimization problems, I cannot find any relevant experiments to verify them.

---

> ### Author Response · Authors · 2024-11-23
> **Rebuttal for Reviewer U39s**
>
> We thank for your positive feedback and the confirmation of our proposed AutoAL! We also thank for your valuable question, so we want to clarify the followings:
>
> **Q1:** My only concern is the efficiency of the AutoAL algorithm. Although more efficient solutions have been proposed to solve second-order optimization problems, I cannot find any relevant experiments to verify them.
>
>    **A1:** The method described in Section 3.3 relaxes the search space, enabling AutoAL to efficiently perform gradient descent for updates, thereby improving its overall efficiency.
>
> During the rebuttal period, we conducted experiments on a relatively large dataset, TinyImageNet. Please refer to Figure 2 in the revised paper. The results demonstrate that AutoAL performs well on large datasets compared to other AL baselines. Additionally, we have analyzed the time cost of AutoAL training. Our AutoAL is efficient, and the primary time consumption part is the AL strategies sampling part but not the AutoAL training itself. The overall time cost of AutoAL is comparable to other baseline methods, such as Ensemble Variance Ratio Learning. Please refer to Section 4.5 for further details.

---

> > ### Comment · Reviewer_U39s · 2024-11-26
> >
> > Thanks for your detailed response. I maintain the original score of 6.

---

### Official Review · Reviewer_164p · 2024-11-02

**Soundness:** 2
**Presentation:** 2
**Contribution:** 2
**Rating:** 3
**Confidence:** 1

**Summary:**

This paper proposes an active strategy query algorithm where the optimal query strategy is selected by a bi-level optimization network. In particular, the authors aggregates the query strategies by a scoring function implemented as a Gaussian Mixture Model. Then, the authors split out a validation set from the labeled samples to guide scoring function calculation. Experimental results show that the proposed method supasses the baselines.

**Strengths:**

1. The studied strategy selection problem for active learning is important.
2. The bi-level optimization strategy is rational.

**Weaknesses:**

1. There is still room for improvement in the paper writing.

   1.1. It's unnecessary to name the two networks "fitnet" and "searchnet," as it seems intended to make people think this is a significant innovation. However, in meta-learning, this kind of separated network design and bi-level optimization paradigm is very common.

   2.1. The notations are somewhat confusing. For example, the authors didn't clearly define the output of the search net in Section 3.2, making it hard to understand Sec 3.2. It wasn’t until I finished reading the method section that I realized the output is actually a sample-wise score, forming an aggregation of scores for different queries.

2. The novelty of this paper is relatively limited. The proposed meta-learning/bi-level optimization has been applied to AL [1,2]. Also, I think the algorithm design is too complicated.

3. The motivation for modeling the scores by GMM distributions is unclear. Why is the score function of each strategy distributed as a Gaussian Distribution? Why is the final score function a linear weighted aggregation of different strategies? The authors should provide a concrete application or example.

4. The comparison methods are too outdated, with the latest ones being LPL and BADGE from 2019. Additionally, the datasets are quite limited; despite the complexity of the method design, only CIFAR and MNIST datasets were used. Validation should be conducted on the ImageNet dataset (at least Image100). Otherwise, given that the algorithm design is much more complex than the baselines, its effectiveness cannot be convincingly demonstrated.

[1] Kunkun Pang, Mingzhi Dong, Yang Wu, and Timothy Hospedales. 2018. Meta-learning transferable active learning policies by deep reinforcement learning. International Workshop on Automatic Machine Learning (ICML AutoML 2018).

[2] https://grlplus.github.io/papers/96.pdf

**Questions:**

see above

---

> ### Author Response · Authors · 2024-11-23
> **Rebuttal for Reviewer 164p**
>
> Thank you for your valuable feedback. We appreciate that you confirm our contribution on solving the strategy selection problem and our rational method: using differentiable bi-level framework.
>
> For the weakness, we make the following comments to clarify our points:
>
> **Q1:** There is still room for improvement in the paper writing.
>    1.1 It's unnecessary to name the two networks "fitnet" and "searchnet," as it seems intended to make people think this is a significant innovation. However, in meta-learning, this kind of separated network design and bi-level optimization paradigm is very common.
>
>    2.1. The notations are somewhat confusing. For example, the authors didn't clearly define the output of the search net in Section 3.2, making it hard to understand Sec 3.2. It wasn’t until I finished reading the method section that I realized the output is actually a sample-wise score, forming an aggregation of scores for different queries.
>
>    **A1:** Thank you for raising these concerns. First, we named the networks "FitNet" and "SearchNet" because we believed these names reflected their tasks: "FitNet" models the data distribution to make AutoAL fit the data, while "SearchNet" searches the best AL strategy in the candidate pool. Thank you for the comment, to align with the net name used in previous AL works [R11], we will rename "FitNet" to "TaskNet" in the final version.
>
>    Second, we apologize for the confusion caused by unclear notations. We have revised the paper to define the outputs of both SearchNet and FitNet explicitly at Section 3.2. Additionally, we will update our main figure to clarify these details for readers.
>
> **Q2:** The novelty of this paper is relatively limited. The proposed meta-learning/bi-level optimization has been applied to AL [R9,R10]. Also, I think the algorithm design is too complicated.
>
>    **A2:** Thank you for referencing these materials. However, we believe the works you mentioned differ significantly from our AutoAL:
>    (1) In [R9], the focus is not on algorithm selection but on treating AL as a meta-learning task using reinforcement learning to predict the next best data point. Additionally, their network is not differentiable, and their use of a reward function results in high computational costs.
>    (2) In [R10], the meta-gradients are computed with respect to perturbations added to labels, rather than through differentiation with model parameters. Furthermore, their approach uses meta-learning as an uncertainty-based active sampler, while AutoAL is designed to select the best strategy from multiple AL candidates. Lastly, their work is targeted at semi-supervised node classification, not image recognition.
>
>    To clarify our contribution: Usually a single AL method cannot perform well across all real-world applications. AutoAL focuses on selecting the best strategy from existing AL methods. Moreover, our network is differentiable, enabling efficient AL strategy selection through gradient descent.
>
> **Q3:** The motivation for modeling the scores by GMM distributions is unclear. Why is the score function of each strategy distributed as a Gaussian Distribution? Why is the final score function a linear weighted aggregation of different strategies? The authors should provide a concrete application or example.
>
>    **A3:** Thank you for raising these insightful questions. First, in AL settings, the strategy will only query a subset of the data. As defined in Section 3.3, the ratio t (batch size b divided by the total pool size M+N) determines the number of data points to query. However, it is difficult to directly identify the top-t data points. To address this, we use a Gaussian Mixture Model (GMM) to model AL scores, setting the t-th best value as the sigmoid zero point. This approach relaxes the search space and effectively supervises the updates to SearchNet using the loss function in Section 3.4.
>
>    Second, the linear aggregation of scores is a design choice tailored to our method. For each image, if a candidate strategy selects it as one of the top data points, it receives a high score (1 in our case); otherwise, it receives a low score (0). The final score for each image is a linear combination of strategy-specific scores weighted by the image's self-loss (computed by FitNet). This design balances the contributions of different strategies and enables effective modeling of final image importance.
>
>    Thank you for the comments, we will add more details on method design in the final paper.

---

> ### Author Response · Authors · 2024-11-23
> **Qestion 4 and References for the Rebuttal for Reviewer 164p**
>
> **Q4:** The comparison methods are too outdated, with the latest ones being LPL and BADGE from 2019. Additionally, the datasets are quite limited; despite the complexity of the method design, only CIFAR and MNIST datasets were used. Validation should be conducted on the ImageNet dataset (at least Image100). Otherwise, given that the algorithm design is much more complex than the baselines, its effectiveness cannot be convincingly demonstrated.
>
>    **A4:** Thank you for the helpful suggestion. We have added TinyImageNet dataset, and, four baselines for comparison: Coreset [ICLR’2018], Ensemble Variance Ratio Learning [CVPR’2018], Variational Adversarial Active Learning [ICCV’2019], Deep Deterministic Uncertainty [CVPR’2023]. The results in Figure 2 demonstrate that our method outperforms all baselines across every dataset, including the newly added TinyImageNet.
>
> Table 1 summarizes the datasets used in this work. We also want to clarify that MedMNIST is not the written number as MNIST, it's a large-scale collection of standardized biomedical images. Such as TissueMNIST, it contains Kidney Cortex Microscope images with total 165466 images as training set.
>
> Reference:
>
> [R9]Pang, Kunkun, et al. "Meta-learning transferable active learning policies by deep reinforcement learning." arXiv preprint arXiv:1806.04798 (2018).
>
> [R10] Madhawa, Kaushalya, and Tsuyoshi Murata. "Active learning on graphs via meta learning." ICML Workshop on Graph Representation Learning and Beyond, ICML. 2020.
>
> [R11] Huang, Siyu, et al. "Semi-supervised active learning with temporal output discrepancy." Proceedings of the IEEE/CVF International Conference on Computer Vision. 2021.

---

> > ### Comment · Reviewer_164p · 2024-11-27
> > **discussion**
> >
> > Thanks for your responses. Still, I think the technical novelty is not as significant as claimed. I decide to maintain my scores.

---

### Official Review · Reviewer_Rqgr · 2024-11-03

**Soundness:** 2
**Presentation:** 2
**Contribution:** 2
**Rating:** 3
**Confidence:** 3

**Summary:**

The paper presents AutoAL, a framework for automated active learning that optimizes query strategy selection using differentiable methods. Traditional active learning approaches often rely on predefined strategies like uncertainty sampling or diversity sampling, which may not perform optimally across different datasets or tasks. AutoAL addresses this limitation by integrating existing active learning strategies into a unified framework. It employs two neural networks, SearchNet and FitNet, within a bi-level optimization structure to automate the selection process. By relaxing the discrete search space of active learning strategies into a continuous domain, AutoAL enables gradient-based optimization, enhancing computational efficiency and adaptability. Experimental results demonstrate that AutoAL consistently outperforms individual strategies and other selective methods across various natural and medical image datasets, highlighting its effectiveness and versatility.

**Strengths:**

- A new approach that automates active learning strategy selection through differentiable optimization, surpassing manual and non-differentiable methods.
- Effective integration of strategy selection and data modeling via the bi-level optimization of SearchNet and FitNet.
- Flexibility and adaptability, allowing incorporation of multiple existing strategies and tailoring to specific tasks and data distributions.

**Weaknesses:**

- Increased complexity and computational overhead due to the additional neural networks and bi-level optimization, potentially challenging scalability on large datasets.
- Dependence on a predefined pool of candidate strategies, which may limit performance if optimal strategies are not included.
- Lack of in-depth theoretical analysis explaining the method's effectiveness and the conditions under which it performs best, possibly affecting generalizability.

**Questions:**

- How does AutoAL perform in terms of computational efficiency compared to traditional methods on large-scale datasets?
- What mechanisms ensure robustness against convergence issues and local minima in the bi-level optimization?
- Can AutoAL be extended to generate new active learning strategies dynamically rather than relying solely on a predefined candidate pool?

---

> ### Author Response · Authors · 2024-11-23
> **Rebuttal for Reviewer Rqgr**
>
> Thank you for your valuable feedback. We appreciate that you confirm our contribution on proposing the first Automatic AL search method with differentiable bi-level framework, which surpasses manual and non-differentiable AL methods.
>
> For your concerns, we make the following comments to clarify our points:
> **Q1:** Increased complexity and computational overhead due to the additional neural networks and bi-level optimization, potentially challenging scalability on large datasets.
>
>    **A1:** We agree that scaling bi-level optimization to large datasets presents efficiency challenges. To address this, the core of our work focuses on designing a differentiable query strategy optimization to significantly reduce computational complexity.
>
> Especially, we have validated AutoAL on two large-scale datasets: TissueMNIST (before rebuttal) with *165,466* images, and TinyImageNet (after rebuttal) with *100,000* images. Our results demonstrate that AutoAL surpasses other baseline models. Additionally, we have included an analysis of computational cost (in terms of time) in the revised paper (see Table 2). The time cost of our AutoAL is compatable to other baselines, such as Ensemble Variance Ratio Learning. We believe these results demonstrate that AutoAL is scalable and generalizable to large datasets.
>
>
> **Q2:** Dependence on a predefined pool of candidate strategies, which may limit performance if optimal strategies are not included.
>
>    **A2:** AutoAL can easily integrate additional AL strategies into the candidate pool. However, even if the optimal AL strategy is not included, our results in Figure 4 show that simply enlarging the candidate pool does not always improve performance. This suggests that AutoAL can still select a relatively effective uncertainty-based or diversity-based strategy from the pool and achieve good performance, even in the absence of the theoretically best strategy.
>
> **Q3:** Lack of in-depth theoretical analysis explaining the method's effectiveness and the conditions under which it performs best, possibly affecting generalizability.
>
>    **A3:** [R5] perform comprehensive theoretical analysis for active learning. As discussed in [R5], AL often performs vary on different problem settings. AutoAL seeks to address this limitation by generalizing to different datasets automatically.
>
> We designed two networks—SearchNet and FitNet—within a bi-level framework. FitNet uses gradient descent to adapt to the labeled data distribution, simulating the final classification model. SearchNet is updated under FitNet's supervision and identifies the most informative data points. This process is supported by the common practice of selecting samples with maximum loss, as discussed in [R6].
>
> From our experiments, AutoAL has consistently outperformed baselines across various datasets, including TinyImageNet (during rebuttal) and additional medical image classification tasks. These datasets are widely used in prior research [R7,R8]. Furthermore, Table 2 in the revised paper highlights AutoAL's efficiency and generalizability. Please refer to Section 4.5 for further details.
>
>
> For the questions, we want to clarify that:
> **Q4:** How does AutoAL perform in terms of computational efficiency compared to traditional methods on large-scale datasets?
>
>    **A4:** We have added the differentiable query strategy optimization (Section 3.3) to relax the search space, which reduces computational complexity. The computational cost analysis included in Table 2 of the revised paper shows that AutoAL's time cost is comparable to methods like ENSvarR and VAAL. Most of the time is spent on the computation of scores of different AL strategies, while the AutoAL network update incurs trivial additional cost. Additionally, the time ratio relative to EntropySampling remains stable even on large-scale datasets, demonstrating that AutoAL scales effectively.
>
> **Q5:** What mechanisms ensure robustness against convergence issues and local minima in the bi-level optimization?
>
>    **A5:** Several mechanisms ensure robustness in our framework:
>
>    (1) FitNet: Convergence is straightforward as FitNet minimizes the loss over the labeled data pool, aligning with the data distribution.
>    (2) SearchNet: We relax the search space, enabling gradient ascent to supervise SearchNet's updates effectively and facilitate convergence.
>    (3) AL character: SearchNet and FitNet are retrained during each AL round using updated data distributions, which allows SearchNet to avoid local minima and converge toward a global solution.

---

> ### Author Response · Authors · 2024-11-23
> **Qestion 6 and References for the Rebuttal for Reviewer Rqgr**
>
> **Q6:** Can AutoAL be extended to generate new active learning strategies dynamically rather than relying solely on a predefined candidate pool?
>
>    **A6:** AutoAL is not designed to generate new active learning strategies dynamically. Instead, it focuses on integrating any kinds of AL strategies into its candidate pool and leveraging them effectively. We believe this approach allows AutoAL to take full advantage of existing advanced strategies while maintaining flexibility and extensibility.
>
> Reference:
> [R5] Mussmann, Stephen O., and Sanjoy Dasgupta. "Constants matter: The performance gains of active learning." International Conference on Machine Learning. PMLR, 2022.
>
> [R6] Huang, Siyu, et al. "Semi-supervised active learning with temporal output discrepancy." Proceedings of the IEEE/CVF International Conference on Computer Vision. 2021.
>
> [R7] Hacohen, Guy, and Daphna Weinshall. "How to select which active learning strategy is best suited for your specific problem and budget." Advances in Neural Information Processing Systems 36 (2024).
>
> [R8] Zhang, Jifan, et al. "Algorithm selection for deep active learning with imbalanced datasets." Advances in Neural Information Processing Systems 36 (2024).

---

> > ### Comment · Reviewer_Rqgr · 2024-11-23
> >
> > I have carefully reviewed the authors' responses. While I fully agree with the motivation and problem statement of AutoAL, I still find the theoretical foundation for the purpose of the bi-level optimizer to be insufficient. The idea of minimizing loss has already been explored in the existing literature, and studies addressing issues like the scaling overhead required for diverse pool selection still seem lacking. I appreciate the effort put into preparing the rebuttal, but I will maintain my original score.

---

### Official Review · Reviewer_fWbJ · 2024-11-11

**Soundness:** 3
**Presentation:** 3
**Contribution:** 2
**Rating:** 6
**Confidence:** 5

**Summary:**

This work introduces AutoAL, a differentiable active learning (AL) strategy search method that builds on existing AL sampling strategies. AutoAL contains two neural networks, SearchNet and FitNet, which are co-optimized through a differentiable bi-level optimization framework to identify optimal AL strategies for different tasks. Experimental results show that AutoAL outperforms individual AL algorithms and other selective approaches.

**Strengths:**

- The work handles an important ML problem; Active Learning (AL) with differentiable strategy search.
- The proposed method is technically sound
- Writing is clear and easy-to-follow

**Weaknesses:**

- Hybrid AL methods that combine uncertainty and diversity have been demonstrated to perform effectively in a variety of situations. It would be beneficial to include examples where the proposed AutoAL approach is particularly necessary or advantageous for specific applications.
- The bi-level optimization within AutoAL relies on labeled data. How does the algorithm perform if the labeled data is skewed or imbalanced? For instance, if the initial labeled set suffers from class imbalance, might this severely impair the algorithm? The assumption of a randomly selected initial set, as used in the current experiments, appears to be less practical.
- Similarly, is there a guarantee that the AutoAL approach, trained with labeled data from the current AL round, will identify the most informative samples from the unlabeled pool in the subsequent AL round? A more detailed analysis of the algorithm's guarantees is necessary.
- The approach of training an additional network for sample selection shows similarities to [1] employing meta-learning with an additional network for querying.
- Can the proposed method be applied to the open-set AL problem [1]?
- The datasets used in the experiments are of small scale. It is imperative to validate the performance on large-scale datasets, such as ImageNet.

---
[1] Meta-Query-Net: Resolving Purity-Informativeness Dilemma in Open-set Active Learning, NeurIPS, 2022

**Questions:**

See weaknesses

---

> ### Author Response · Authors · 2024-11-23
> **Rebuttal for Reviewer fWbJ**
>
> Thank you for your valuable comments. We appreciate that you confirm our contribution on proposing the first Automatic AL search method with a differentiable bi-level framework, which will make AL automatically suitable to different real-life applications.
>
> For your questions, we make the following comments to clarify our points:
> **Q1:** Provide examples where the proposed AutoAL approach is particularly advantageous compared to hybrid AL methods that combine uncertainty and diversity
>
>    **A1:** While hybrid methods have shown relatively good performance in some real-world applications compared to single uncertainty- or diversity-based methods, determining the optimal trade-off between these strategies remains a challenge. This limitation makes hybrid methods less robust across all scenarios because they often rely on fixed heuristics, such as weighted-sum [R2] or multi-stage optimization [R3]. For example, in our revised paper Figure 2, BadgeSampling [R3] can outperform single uncertainty- or diversity-based methods in some cases (e.g., SVHN), but it performs poorly in others, such as PathMNIST.
>
>    In contrast, AutoAL treats the trade-off between various AL candidates as a learning task. It is the first method capable of automatic selection among candidate AL strategies. Our experimental results demonstrate that AutoAL generalizes well to diverse real-world applications, including both natural and medical image datasets.
>
>
> **Q2:** How does AutoAL handle skewed or imbalanced labeled data, particularly if the initial labeled set suffers from class imbalance? Would this impair its performance, given the assumption of a randomly selected initial set?
>
>    **A2:** Thank you for raising this insightful question. Our method is not specifically designed to address the class imbalance problem. While, we have conducted experiments on imbalanced datasets, such as the medical datasets (see Table 1 for detailed descriptions of these datasets). Additionally, to ensure AutoAL's robustness in such scenarios, we repeated our trials three times.
>
>    The experimental results indicate that AutoAL exhibits very low standard deviation, demonstrating that even when the initial selection pool is not carefully balanced, AutoAL consistently selects the best strategy and outperforms other baselines.
>
>
> **Q3:** Does AutoAL guarantee that training with labeled data from the current AL round will identify the most informative samples in the next round? A detailed analysis of its guarantees would be helpful.
>
>    **A3:** AutoAL samples data uniformly from the pool, ensuring that the data in each cycle follows approximately the same distribution. The experimental results confirm that this approach works well, as AutoAL consistently outperforms baselines in every AL round. Furthermore, AutoAL shows minimal accuracy drops between rounds. Some baseline methods, for instance, MarginSampling on SVHN, suffers a severe accuracy drop in round 4. In addition, this sampling approach is a standard practice in learning-based AL methods [R4] to ensure a good generalization ability across AL rounds.
>
>
> **Q4:** The approach of training an additional network for sample selection shows similarities to [R1] employing meta-learning with an additional network for querying.
>
>    **A4:** Thank you for mentioning this meta-learning work. We believe AutoAL has significant differences from MQNet:
> (1) MQNet focuses on training an MLP that takes an open-set score and an AL score as input, outputting a balanced meta-score for sample selection. In contrast, AutoAL trains an active learning selection network to determine the most effective AL strategy based on the labeled dataset;
> (2) To address efficiency concerns, AutoAL employs a bi-level optimization strategy and differentiable query strategy optimization (see Sec. 3.3), which is not mentioned by MQNet;
> (3) MQNet is designed for open-set dilemma, whereas AutoAL focuses on datasets where class labels are present in the labeled set.
> 	Thank you for the helpful suggestion, we have revised our Related Work accordingly.
>
> **Q5:** Can the proposed method be applied to the open-set AL problem [R1]?
>
>    **A5:** As mentioned in **Q4**, AutoAL is not designed to address the open-set problem. However, it can automatically balance uncertainty and diversity within in-distribution (IN) data. But we really thank for the insightful advice and we will consider this as the future work of AutoAL.
>
> **Q6:** The datasets used in the experiments are of small scale. It is imperative to validate the performance on large-scale datasets, such as ImageNet.
>
>    **A6:** We have added experiments on the TinyImageNet dataset (200 classes). Please refer to Figure 2 in the revised paper. Our method outperforms all baselines at every round, demonstrating the scalability of AutoAL.

---

> ### Author Response · Authors · 2024-11-23
> **References for the Rebuttal for Reviewer fWbJ**
>
> References:
> [R1] Park, Dongmin, et al. "Meta-query-net: Resolving purity-informativeness dilemma in open-set active learning." Advances in Neural Information Processing Systems 35 (2022): 31416-31429.
>
> [R2] Yin, Changchang, et al. "Deep similarity-based batch mode active learning with exploration-exploitation." 2017 IEEE international conference on data mining (ICDM). IEEE, 2017.
>
> [R3] Ash, Jordan T., et al. "Deep Batch Active Learning by Diverse, Uncertain Gradient Lower Bounds." International Conference on Learning Representations (ICLR). 2020.
>
> [R4] Clarysse, Jacob, and Fanny Yang. "Uniform versus uncertainty sampling: When being active is less efficient than staying passive."

---

> > ### Comment · Reviewer_fWbJ · 2024-11-25
> >
> > Thanks for the clarification. I have read all the responses. I would like to keep my original score.

---

### Author Response · Authors · 2024-11-23
**For All the Reviewers, PCs, ACs**

We apologize for the delayed response. We sincerely thank all the reviewers for their valuable feedback and the time dedicated to improving our work. We have carefully reviewed all comments and made the following changes to our paper during the rebuttal period:

1. We added **four baselines**: Coreset [ICLR 2018], Ensemble Variance Ratio Learning [CVPR 2018], Variational Adversarial Active Learning (VAAL) [ICCV 2019], and Deep Deterministic Uncertainty (DDU) [CVPR 2023], along with the **TinyImageNet dataset** to verify the generalizability of AutoAL. These new results further demonstrate that AutoAL outperforms all baselines across different settings. Please refer to the updated Figure 2 and Section 4 in the revised paper for more details.
2. We included a comprehensive **method efficiency analysis** of AutoAL and other baselines on different datasets in Table 2. The results indicate that the complexity and time cost of AutoAL are comparable to other baselines, such as Ensemble Variance Ratio Learning. Moreover, we found that the most time-consuming component is AL strategy sampling, not the AutoAL update process. This suggests that even with bi-level optimization, the relaxed search space and differentiable design of AutoAL ensure that it does not significantly increase time cost.
3. We clarified the differences between our work and meta-learning in the revised Related Work section.

Additionally, we would like to highlight the contribution of this work again:
1. AutoAL is specifically designed for automatic active learning strategy selection in diverse real-world settings. Our bi-level optimization framework allows integration of *any existing AL methods* into the candidate pool, enabling the selection of the most effective strategy for a given task.
2. AutoAL is *the first differentiable AL strategy selection method*. Key innovations, such as the probabilistic query strategy and differentiable acquisition function optimization, effectively reduce the complexity of bi-level optimization, making the time cost of AutoAL comparable to traditional AL strategies.
3. Our experimental results demonstrate that AutoAL achieves state-of-the-art performance across all datasets, including the TinyImageNet dataset. These findings highlight the generalizability of our proposed method in real-world applications.

We hope you could spend some time to review our revised paper and our responses to your questions. If you have any further questions, we would be more than happy to provide additional clarifications by the rebuttal deadline date.

---

### Meta-Review · Area_Chair_Wx15 · 2024-12-20

**Metareview:**

This work introduces AutoAL, the first differentiable active learning strategy search method, which uses a bi-level optimization framework to adaptively identify optimal AL strategies, achieving superior accuracy and efficiency across diverse tasks and domains.

There reviewers reviewed this paper. All of them agree that this paper is not yet ready for publication at this stage. We recommend the authors go through all the reviews and address it in the next version.

**Additional Comments On Reviewer Discussion:**

There reviewers reviewed this paper. All of them agree that this paper is not yet ready for publication at this stage. We recommend the authors go through all the reviews and address it in the next version.

---

### Decision · Program_Chairs · 2025-01-22

Reject